# Immunomodulatory potential of *Clinacanthus nutans* extracts in the co-culture of triple-negative breast cancer cells, MDA-MB-231, and THP-1 macrophages

Fariza Juliana Nordin[1], Lishantini Pearanpan[1], Kok Meng Chan[2], Endang Kumolosasi[3], Yoke Keong Yong[4], Khozirah Shaari[5], Nor Fadilah Rajab[1]*

1 Biomedical Science Program, Center for Healthy Aging and Wellness, Faculty of Health Sciences, Universiti Kebangsaan Malaysia, Kuala Lumpur, Malaysia, 2 Center for Toxicology and Health Risk Studies, Faculty of Health Sciences, Universiti Kebangsaan Malaysia, Kuala Lumpur, Malaysia, 3 Drug and Herbal Research Centre, Faculty of Pharmacy, Universiti Kebangsaan Malaysia, Kuala Lumpur, Malaysia, 4 Department of Human Anatomy, Faculty of Medicine and Health Sciences, Universiti Putra Malaysia, UPM Serdang, Seri Kembangan, Selangor, Malaysia, 5 Faculty of Science, Universiti Putra Malaysia, UPM Serdang, Seri Kembangan, Malaysia

* nfadilah@ukm.edu.my

**Data Availability Statement:** All supporting information files are available from the Figshare

## Abstract

Triple-negative breast cancer is the main type of breast carcinoma that causes mortality among women because of the limited treatment options and high recurrence. Chronic inflammation has been linked with the tumor microenvironment (TME) in breast cancer progression. *Clinacanthus nutans* (CN) has gained much attention because of its anticancer properties, but its mechanism remains unclear. We aimed to study the qualitative phytochemical content and elucidate the cytotoxicity effects of CN on human triple-negative breast cancer (TNBC), MDA-MB-231 and human macrophage-like cells such as THP-1 by using sulforhodamine B (SRB) assay. As highly metastatic cells, MDA-MB-231 cells can migrate to the distal position, the effect of CN on migration were also elucidated using the scratch assay. The CN effects on ameliorating chronic inflammation in TME were studied following the co-culture of MDA-MB-231/THP-1 macrophages. The cytokine expression levels of IL-6, IL-1β and tumor necrosis factor-alpha (TNF-α) were determined using ELISA assays. The results showed that both ethanolic and aqueous CN extracts contained alkaloid, phenol and tannin, flavonoid, terpenoid, glycoside and steroid. However, saponin was only found in the aqueous extract of CN. CN was not cytotoxic to both MDA-MB-231 and THP-1 cells. The ability of MDA-MB-231 to migrate was also not halted by CN treatment. However, CN ethanol extract decreased IL-6 at 25 μg/mL ($p = 0.02$) and 100 μg/mL ($p = 0.03$) but CN aqueous extract increased IL-6 expression at 50 μg/mL ($p = 0.08$) and 100 μg/mL ($p = 0.02$). IL-1β showed decreased expression after treated with CN ethanol and CN aqueous both at 25 μg/mL ($p = 0.03$). TNF-α were significantly decreased after CN ethanol treatment at concentration 25- ($p = 0.001$), 50- ($p = 0.000$) and 100 μg/mL ($p = 0.000$). CN aqueous extract slightly inhibited TNF-α at all 25–50- and 100 μg/mL ($p = 0.001$, $p = 0.000$, $p = 0.000$, respectively). Overall, CN acts by ameliorating the pro-inflammatory condition in

database https://doi.org/10.6084/m9.figshare.14212250.v2

**Funding:** This research was financially supported by NKRA Research Grant Scheme (NRGS) from Ministry of Agriculture (Grant number NH1014D071) URL: https://www.mafi.gov.my/. The funder had no role in study design, data collection and analysis, decision to publish, or preparation of the manuscript.

**Competing interests:** The authors have declared that no competing interests exist.

**Abbreviations:** CN, *Clinacanthus* nutans; DC, dendritic cell; DMSO, dimethyl sulfoxide; ECM, extracellular matrix; ER, estrogen receptor; HEPES, 4-(2-hydroxyethyl)-1-piperazineethanesulfonic acid; HER2, human epidermal growth factor receptor 2; HPLC, high-performance liquid chromatography; $IC_{50}$, inhibition concentration 50; IL-1β, interleukin-1 beta; IL-6, interleukin-6; IL-8, interleukin-8; Jak, janus kinase; LPS, lipopolysaccharide; MDSC, myeloid-derived suppressor cell; MMP, matrix metalloproteinase; NF-κB, nuclear factor kappa B; NK, natural killer; PD-L1, programmed death-ligand 1; PMA, phorbol 12-myristate 13-acetate; PR, progesterone receptor; SRB, sulforhodamine B; STAT, signal transducer and activator of transcription; TAM, tumor associated macrophage; TCA, trichloroacetic acid; TLR-4, toll-like receptor-4; TME, tumor microenvironment; TNBC, triple-negative breast cancer; TNF-α, tumor necrosis factor alpha; USA FDA, United States of America Food and Drug Association.

the TME and may be a potential strategy for its anticancer mechanism on highly metastatic breast cancer condition. The major pathways that link both cancer and inflammation were NF-κB and STATs thus further study on the upstream and downstream pathways is needed to fully understand the mechanism of CN extracts in cooling the inflamed TME in breast cancer.

## Introduction

Breast cancer is the leading cause of death among women in the world, in which triple-negative breast cancer (TNBC) accounts for approximately 15%–20% of all breast carcinoma [1]. TNBC is characterized by the negative expression of the three main breast cancer biomarkers, namely, estrogen (ER), progesterone (PR) and epidermal-growth-factor-2 (HER2) receptors. This condition is associated with rapid growth, distant metastasis and poor prognosis compared with other breast cancer subtypes [2]. The lack of treatment options for TNBC limits the management of this disease [1].

The tumor microenvironment (TME) is composed of the extracellular matrix (ECM) and numerous types of stromal cells, such as endothelial and immune cells, fibroblasts and adipocytes, which play important roles in tumor progression and the response to treatment [3]. The interaction between cancer and heterogenous cells within the TME is critical for carcinogenesis, because TME is an important facilitator of immune escape and cancer progression [4]. During the transition of in situ to invasive carcinoma, tumor and stromal cells secrete ECM-degrading proteases such as matrix metalloproteinases (MMPs), thus destructing the ECM. ECM degradation enables the tumor cells to invade locally and produce aberrantly secreted proteolytic enzymes, chemokines and cytokines to attract leukocytes, modulate tumor remodeling and increase tumor cell invasion to distant organs, leading to metastasis [3, 5]. Targeting the TME, particularly the immune cells, may revert the immune system into a more anti-tumor state [6] that could be beneficial to patients with TNBC setting.

*Clinacanthus nutans* (CN) (Acanthaceae), which is locally known as "Belalai Gajah" or Sabah snake grass, has gained much attention because of its anticancer properties [7, 8]. As anticancer remedies, the leaves of this plant are commonly used as water decoction for oral ingestion [9]. CN has been studied for its cytotoxic effect against various cancer cell lines. CN induces significant cell death in some types of cancer cells [7] but not in others [10, 11]. The immunomodulatory effects of CN are related to toll-like receptor-4 (TLR-4) and the reduced cytokine secretion in murine macrophage cells [10]. However, this receptor is not involved in the mechanism of apoptotic cell death induced by the combination of CN with gemcitabine in pancreatic cancer cells [11]. This study aimed to investigate the effect of CN on human metastatic breast cancer cells such as MDA-MB-231 alone and in co-culture with human macrophage-like cells such as THP-1. We hypothesized that CN could inhibit breast cancer progression by ameliorating the TME, especially by reducing the pro-inflammatory cytokine expression between cancer and immune cells.

## Materials and methods

### Chemicals and reagents

Undenatured ethanol, chloroform and glacial acetic acid were purchased from HmbG chemicals, Hamburg, Germany. Iodine potassium iodide, ferric chloride, sodium hydroxide,

sulphuric acid, 4-(2-hydroxyethyl)-1-piperazineethanesulfonic acid (HEPES), β-mercap-toethanol in PBS, sullforhodamine B (SRB), phorbol 12-myristate 13-acetate (PMA), trichloro-acetic acid (TCA), Trizma base, bacterial lipopolysaccharides (LPS), and dimethyl sulfoxide (DMSO) were purchased from Sigma-Aldrich, USA. RPMI 1640 (Gibco), 1X antibiotic (Gibco), and foetal bovine serum (FBS) were purchased from Thermo Fischer Scientific, USA.

## Plant extraction

CN plant was collected from the TKC Herbal Nursery in Jelebu, Negeri Sembilan, Malaysia (Coordinates: 2.770028, 101.995321). The plant sample was sent to Herbarium Unit, Universiti Kebangsaan Malaysia (UKM) for identification (voucher specimen no: ID029/2020). The bota-nist, Dr. Shamsul Khamis used microscopy technique for plant identification.

CN leaves extracts were obtained as previously described [8, 12]. Briefly, the plant was har-vested and cleaned from dirt and soil by rinsing with tap water. The leaves were separated from the stem and air-dried at room temperature until fully dried. The dried leaves were milled into powder by using a grinder. Uniform particle size of the leaves powder was obtained using 315 mm test sieve (Retsch, Germany). CN leaves powder were kept in an air-tight con-tainer in cool and dry place at room temperature for storage.

Extraction was done using maceration technique by which powdered CN leaves was soaked in 100% undenatured ethanol or distilled water at a ratio of 1:50 for three days with frequent agitation in an amber container kept away from light (Fig 1). Then, the extracts were filtered before repeating the maceration process with fresh solvent until the filtrates turns light green. The filtrates were combined and concentrated under reduced pressure drying using a rotary evaporator (Heidolph, Germany) with temperature set at 40˚C. The extracts were collected in glass sample container and further dried in fume hood. When the extracts reach constant weight, the yield percentage was calculated using the formula: weight of dried extract (g)/ weight of leaves powder (g) x 100. The extracts were kept at 4˚C until use.

## Qualitative phytochemistry analysis

Qualitative phytochemistry analysis was carried out according to previous studies [13, 14]. All extracts were prepared by stirring 0.5 g of CN extracts in 5 mL of distilled water and then fil-tered using Whatman filter paper.

**Saponin test.**  CN extracts were dissolved in distilled water and shaken vigorously. Stable foam formation indicated the presence of saponin.

**Alkaloid test.**  Alkaloid compounds were determined using Wagner's test. Approximately 2 g of iodine granules and 1.27 g of potassium iodide was mixed with 5 mL of distilled water. After the solution was homogenous, distilled water was added to make 100 mL of the solution. A few drops of the Wagner's solution were added into the CN extracts. The formation of brown or reddish precipitation indicated the presence of alkaloid compounds.

**Phenol and tannin test.**  Two mL of 2% (*v*/*v*) ferric chloride solution was mixed with 10 mL of CN extracts. The presence of phenol and tannin was confirmed when the mixture turned from blue-green to black.

**Flavonoid test.**  Two mL of 2% (*v*/*v*) sodium hydroxide was added to 10 mL of CN extracts, followed by a few drops of 10% (*v*/*v*) sulphuric acid. The transformation from an intense yellow color to colorless indicated the presence of flavonoid compounds.

**Terpenoid test.**  Two mL of chloroform solution was added into 10 mL of CN extracts. Three mL of concentrated sulphuric acid was added carefully to form a layer. Reddish brown coloration at the interface indicated the presence of terpenoid compounds.

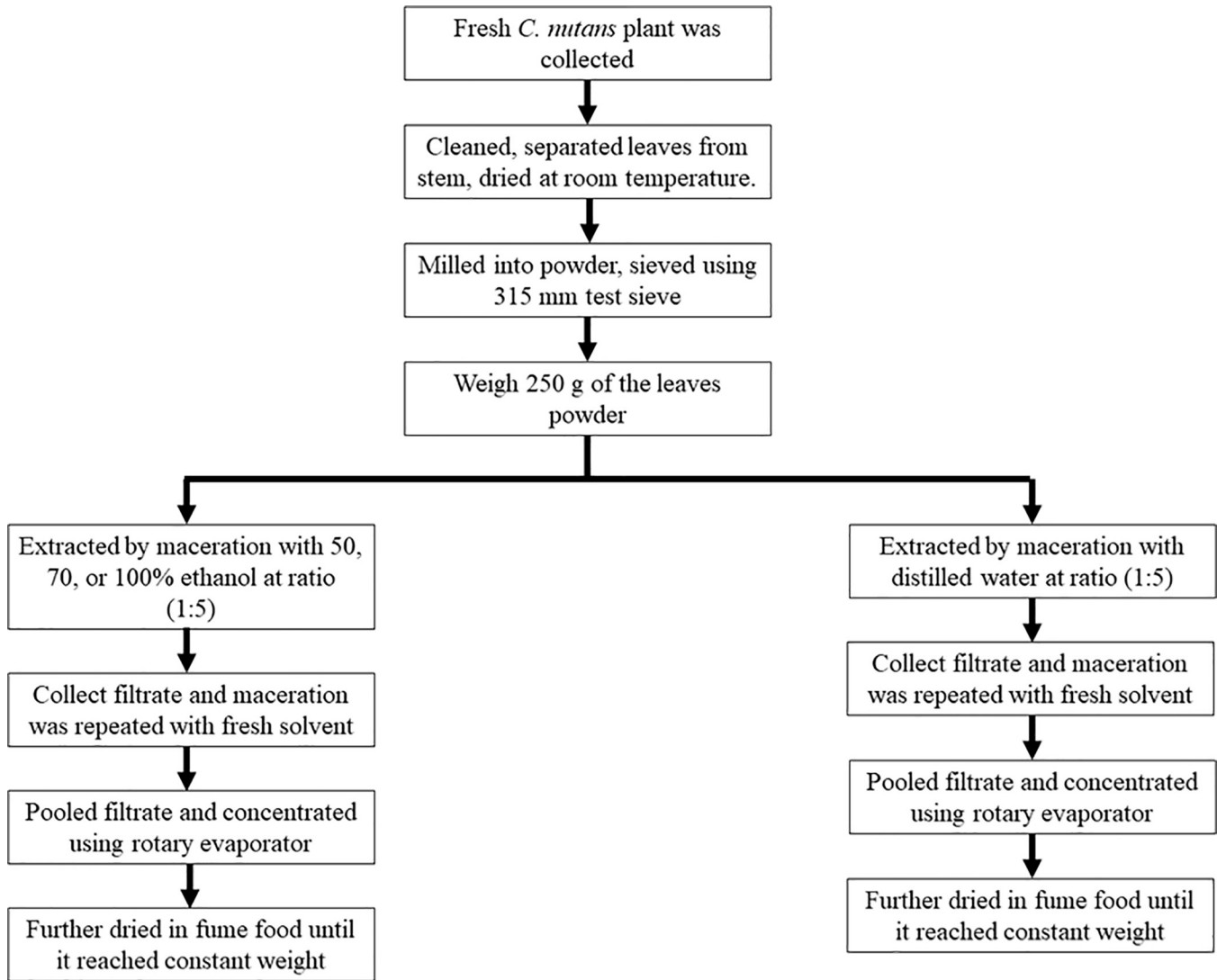

**Fig 1. CN extraction flow chart.**

**Glycosides test.** Keller-Kiliani test were carried out to determine the presence of glycosides in CN extracts. Approximately, 5 mg of CN extract were dissolved in 1 mL of glacial acetic acid. A few drops of ferric chloride solution were added. Then, 2 mL of concentrated sulphuric acid were added slowly into the mixture. Reddish-brown layer was formed in between of extract and the sulphuric acid layers. The upper part of the mixture turns bluish green indicates the presence of glycosides.

**Steroid test.** Two mL of chloroform and sulphuric acid were added to 5 mL of CN extracts on the sidewise of the test tube. Formation of reddish-brown ring in between of chloroform and suphuric acid layers indicates the presence of steroid in the extracts.

## HPLC profiling

CN ethanol and aqueous extracts were sent to Forest Research Institute of Malaysia (FRIM) for HPLC profiling. Briefly, the standard compounds which were schaftoside, isoorientin, orientin, isovitexin and vitexin were diluted in methanol to make 1000 μg/mL of stock

**Table 1. Gradient solvent system.**

| Time (min) | % A | % B |
|:---:|:---:|:---:|
| | **(0.1% formic acid in distilled water)** | **(Acetonitrile)** |
| 0 | 95.0 | 5.0 |
| 30 | 81.0 | 19.0 |
| 33 | 5.0 | 95.0 |
| 38 | 5.0 | 95.0 |
| 39 | 95.0 | 5.0 |
| 42 | 95.0 | 5.0 |

solution, respectively. Serial dilution of each standard compounds was prepared at concentration range of 31.25–500 µg/mL. Approximately 100 mg of CN extracts were dissolved in 5 mL of methanol and sonicated for 20 minutes. Then, the extracts were filtered using PTFE 0.45 µm syringe filter. The extracts were analyzed using HPLC (WATERS 600 quarternary gradient pump, WATERS 717plus autosampler and WATERS 2996 PDA) and HPLC Kinetex Biphenyl C18 (5µm, 250 mm x 4.6mm). The solvent system used were depicted in Table 1. The flow rate was 1.0 mL/min and the injection volume was 10 µL. The detection and quantification of schaftoside, isoorientin, orientin, isovitexin and vitexin was carried out at 330 nm. The retention time values obtained from the selected standard compounds were presented in Table 2.

## Cell culture

MDA-MB-231 (ATCC HTB-26) was obtained from American Type Culture Collection (ATCC; Manassas, VA, USA) and cultured in RPMI 1640 (Gibco, Carlsbad, CA) supplemented with 1X antibiotic (Gibco, Carlsbad, CA), 25 mM HEPES and 5% foetal bovine serum. THP-1 (ATCC TIB-202) monocyte cells were maintained in the same supplemented medium as MDA-MB-231 but with addition of 0.05 mM β-mercaptoethanol in PBS (Sigma-Aldrich, USA). All cells were maintained in an incubator humidified with 5% $CO_2$ at 37˚C.

## Sulforhodamine (SRB) cytotoxicity assay

SRB assay was carried out as previously described to determine the 50% inhibition concentration ($IC_{50}$) values of all the compounds [15, 16]. Initially, the cells were seeded in 96-well plates at a density of 2 x $10^5$ cells/mL (MDA-MB-231), followed by incubation overnight to allow the cells to adhere to the bottom of the plates. The MDA-MB-231 cells were treated with CN 50%, 70%, 100% ethanolic or aqueous extracts at the concentration range of 0–2.0 mg/mL in separate 96-well plate, THP-1 monocyte cells, 2 x $10^5$ cells/mL were seeded and treated with 100 ng/ml phorbol 12-myristate 13-acetate (PMA) (Sigma-Aldrich, USA) to induce the differentiation of monocytes into macrophage-like cells. After 5 days, THP-1 macrophages were treated

**Table 2. The name of standards and their retention time values.**

| Name of standard compound | Retention time |
|:---:|:---:|
| Schaftoside | 29.990 |
| Isoorientin | 30.150 |
| Orientin | 31.351 |
| Isovitexin | 34.682 |
| Vitexin | 34.761 |

with CN 50%, 70%, 100% ethanolic or aqueous extracts at concentration range of 0–2.0 mg/mL. After 48 h of CN treatment, the cells were fixed in the plates by using 50 μL of 50% (*w/v*) trichloroacetic acid solution (Sigma-Aldrich, USA), followed by incubation at 4˚C for 1 h. The plates were then washed for five times with tap water and air-dried prior to staining with 100 μL of 0.4% (*w/v*) SRB staining solution. Further incubation was done for 10 min at room temperature. Subsequently, the plates were washed for three times with 1% (*v/v*) acetic acid to remove the unbound stains. After air-drying, the wells were added with 200 μL of 10 mM Trizma base (Sigma-Aldrich, USA) and shaken well for 10 min. Finally, the absorbance was measured using iMark™ microplate reader (Bio-Rad Laboratories, Hercules, CA, USA) at a wavelength of 490 nm. All experiments were carried out in triplicates.

## Migration assay

Migration assay were carried out according to previous study [17]. In this experiment, MDA-MB-231 cells were seeded in six-well plates at 1 x $10^7$ cells/well. Straight lines were created using a sterile 200 μL pipette tip once the cells reached 100% confluency. The plate was rinsed with sterile PBS to remove floating cells. Cells were then exposed to 0–2 mg/mL CN extracts diluted in serum-free RPMI 1640 media. The wound conditions were monitored for 24 h. Images were captured using Olympus CKX41 light microscope at 20X magnification.

## Co-culture of MDAMB-231/THP-1 macrophage-like cells

Co-culture of MDA-MB-231/THP-1 macrophage-like cells were carried out according to previous studies (Fig 2) [10, 18]. THP-1 monocytes were seeded in the Falcon™ cell inserts (1 μm pore size) at the concentration of 2 x $10^5$ cells/mL. Approximately 100 ng/mL of PMA solution was added into the media and incubated for 5 days to differentiate the monocytes into macrophages. MDA-MB-231 cells were seeded in the Falcon™ companion plate a day before THP-1 incubation with PMA ended. On the fifth day, the THP-1 macrophages and MDA-MB-231 cells were co-cultured in a plate and then treated with CN ethanol or aqueous extracts in serum-free media for 1 h, followed by exposure to 20 ng/mL LPS. Plates were further incubated for 18 h at 37˚C in an incubator humidified with 5% $CO_2$. The media from upper and lower chambers were collected, pooled and concentrated using Amicon Ultra-4 (Millipore) for the IL-6, IL-1β and tumor necrosis factor alpha (TNF-α) cytokine assay.

## Cytokine analysis

The secreted IL-6 IL-1β and TNF-α in the culture medium were quantitated using specific cytokine ELISA kits for IL-1β (Cat no. EZHIL1β), IL-6 (Cat no. EZHIL6), and TNF-α (Cat no. EZHTNFA) (Merck, MA, USA) according to the manufacturer's instructions. Briefly, the plate was washed 4 times with 300 μL/well of 1X Wash Buffer. Residual buffer was blotted firmly by tapping the plate upside down on absorbent paper. Approximately, 50 μL of Assay Buffer A was added to all wells. Then, 50 μL of standard protein or samples were added to the appropriate wells. The plate was sealed and incubated at room temperature for 2 hours. After 2 hours, the plate was washed 4 times with wash buffer. The detection antibody solution was added into the well and further incubate for an hour at room temperature. Then, the plate was washed 4 times using wash buffer. The wells were soaked with wash buffer for 30 second during the final wash to minimize the background. Hundred microliters of Substrate Solution F provided in the kit were added into each well and incubated for 15 minutes in the dark. The solution color changed from blue to yellow. Optical density was measured using a microplate reader at 450 and 570 nm.

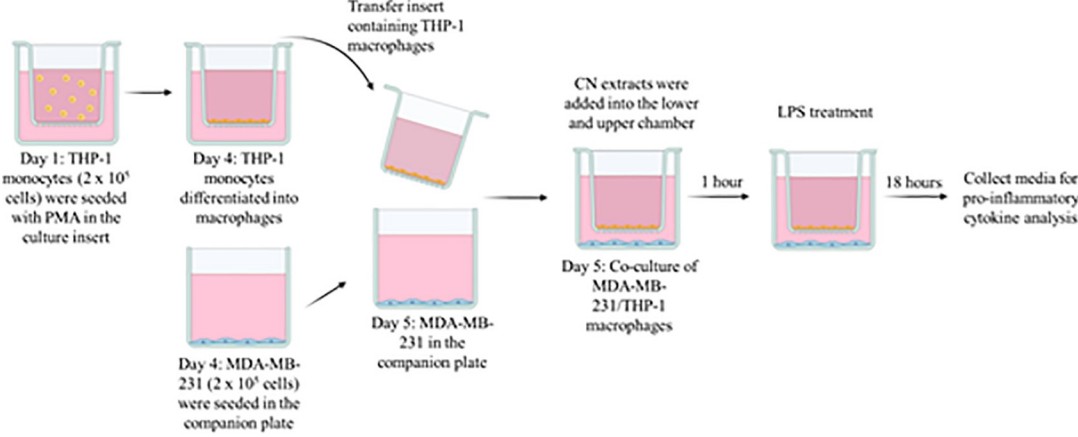

**Fig 2. Schematic diagram of the co-culture experiment.**

## Statistical analysis

All data were analyzed using SPSS version 20 and were presented as the mean ± SD. All data were checked for normality using Shapiro-Wilks's test. One-way ANOVA was applied with Tukey's post hoc test. However, for data that violated the homogeneity of variance assumption, Welch statistical analysis was done with Games-Howell post hoc test. The differences between experimental groups were considered statistically significant at $^*p \leq 0.05$, $^{**}p < 0.01$, $^{***}p < 0.001$, or $^{****}p < 0.0001$.

## Result

### CN qualitative phytochemistry analysis

Qualitative phytochemistry analysis was carried out on both ethanolic and aqueous CN extracts. The result showed that both ethanolic and aqueous CN extracts indicated the presence of alkaloid, phenol and tannin, flavonoid, terpenoid, glycoside, and steroid compounds. However, only aqueous CN extract contained saponin, as shown in Table 3.

CN extraction yields were calculated as below:

Yield CN ethanol (%) = Extract dry weight (g)/Initial extract weight (g) x 100

$$= 22.5057g/250g \times 100$$
$$= 9.0\%$$

**Table 3. CN ethanol and aqueous qualitative phytochemistry analysis.**

| Compound | *Clinacanthus nutans* extracts | |
|---|---|---|
|  | **Ethanol** | **Aqueous** |
| Saponin | - | + |
| Alkaloid | + | + |
| Phenol and Tannin | + | + |
| Flavonoid | + | + |
| Terpenoid | + | + |
| Glycoside | + | + |
| Steroid | + | + |

+Present; -Absent.

Yield CN aqueous = Extract dry weight (g)/Initial extract weight (g) x 100

= 63.531g/250g X 100

= 25.4%

## HPLC profiling

Reference standard compounds schaftoside, isoorientin, orientin, isovitexin and vitexin were prepared in methanol. All reference standard compounds were detected in the chromatogram at 330 nm (Fig 3). HPLC profiling was carried out for both CN ethanol and aqueous extract. From the chromatogram, all compounds were presents in CN ethanol such as schaftoside, iso-orientin, orientin, isovitexin and vitexin (Fig 4A). However, these compounds were not detected in aqueous extract (Fig 4B). Schaftoside was the most abundant compound in CN ethanol extract but not in CN aqueous extract (Table 4).

## Cytotoxic effect of CN on MDA-MB-231 metastatic breast cancer cell lines and THP-1 macrophages

One of the most fundamental steps in anticancer drug discovery is the screening of the anti-proliferative and cytotoxicity activity against cancer cells. The present study showed no $IC_{50}$, as determined using the SRB assay, indicating that CN does not inhibit half of the viability of human metastatic breast cancer cells, MDA-MB-231 and human macrophage-like THP-1 cells. (Fig 5A and 5B). Both MDA-MB-231 cells and THP-1 macrophage-like cells were treated with both ethanol and aqueous extracts for up to 48 h.

## Effect of CN on MDA-MB-231 metastatic breast cancer cell migration

The ability of breast cancer cells to migrate enables them to undergo metastasis to secondary sites, such as bones and lungs. Considering that CN inhibits the metastasis of cancer cells to other organs, we evaluated the effectiveness of CN in inhibiting the migration of highly metastatic breast cancer cells such as MDA-MB-231. These cells successfully migrated into the scratched areas and were not inhibited by CN treatment up to the highest tested concentration of 2 mg/mL following 24 h of exposure (Fig 6).

## Effect of CN on IL-6, IL-1β and TNF-α expression levels

Cytokines are small secreted proteins released by cells that have a specific effect on the interactions and communications between cells. CN 100% ethanol and aqueous extracts were used to investigate the cytokine expression in the media of the co-culture of MDA-MB-231 and THP-1 macrophage-like cells. The present study showed that CN extracts may help in ameliorating the inflammation state between human breast cancer cells, MDA-MB-231 and THP-1 macro-phage-like cells. The expression of IL-6 cytokine in the media of MDA-MB-231/THP-1 macro-phage co-culture was significantly reduced at 25 and 100 μg/mL but not at 50 μg/mL compared with LPS-treated control (Fig 7A). IL-1β cytokine expression was significantly reduced at 25 μg/mL for both CN ethanol and aqueous (Fig 7B). Moreover, TNF-α expression was significantly reduced after treatment with CN ethanol and aqueous when compared to LPS-treated group. (Fig 7C).

## Discussion

Triple-negative breast cancer (TNBC) account for 10%–20% of all types of breast cancer. Inci-dentally, this minority type of breast cancer accounts for many recurrent and metastatic cases and breast cancer deaths [19]. The only therapy option for patients with TNBC is cytotoxic

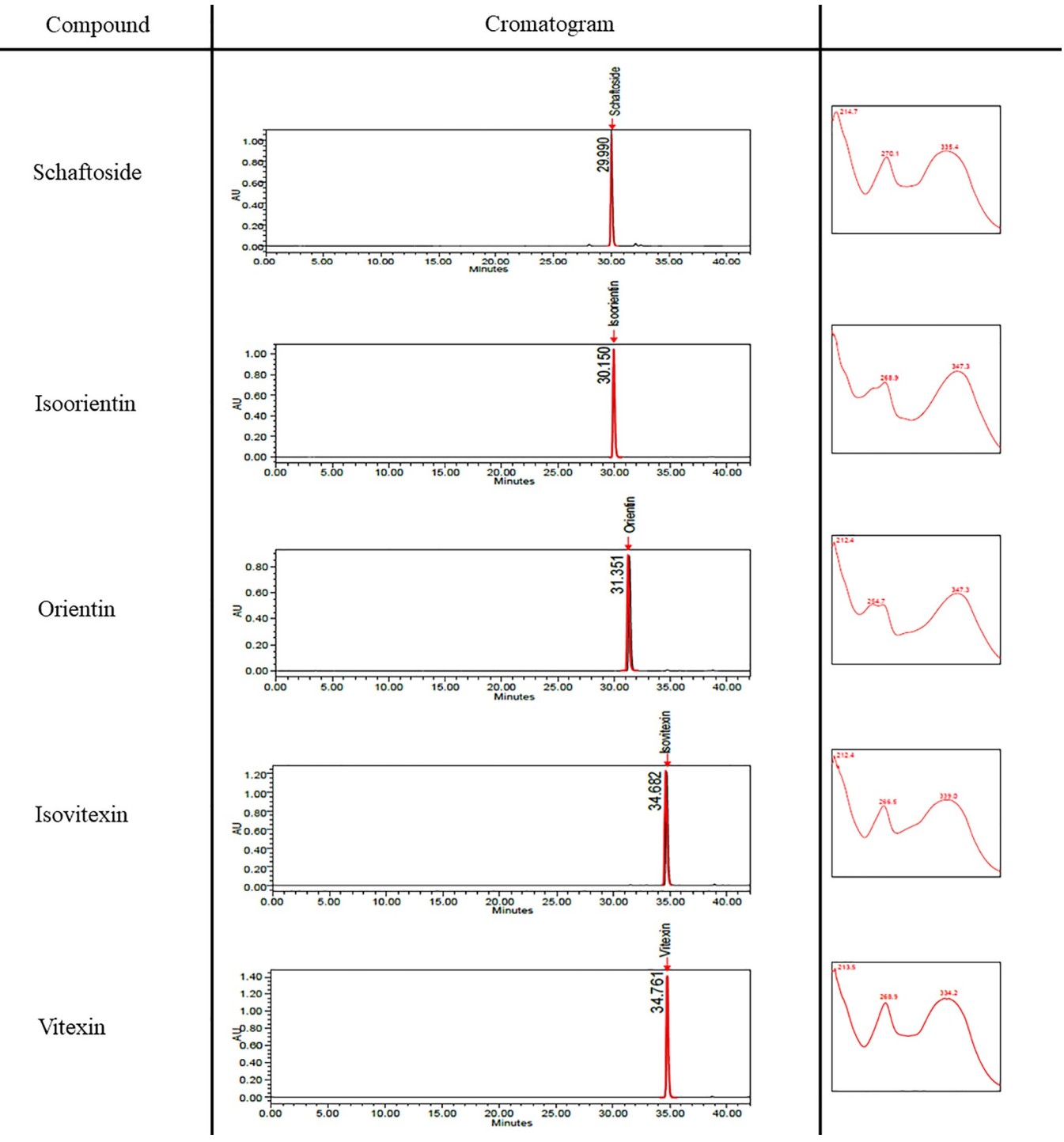

**Fig 3. HPLC chromatogram of the reference standard compounds.**

chemotherapy [20]. Recently, IMpassion130 trial has brought atezolizumab as the pioneer immunotherapy agent for TNBC, thus changing the cancer treatment paradigm [21]. However, this USA Food and Drug Association (FDA)-approved agent has only been applied to patients with TNBC whose breast cancer expresses programmed death-ligand 1 (PD-L1) [22].

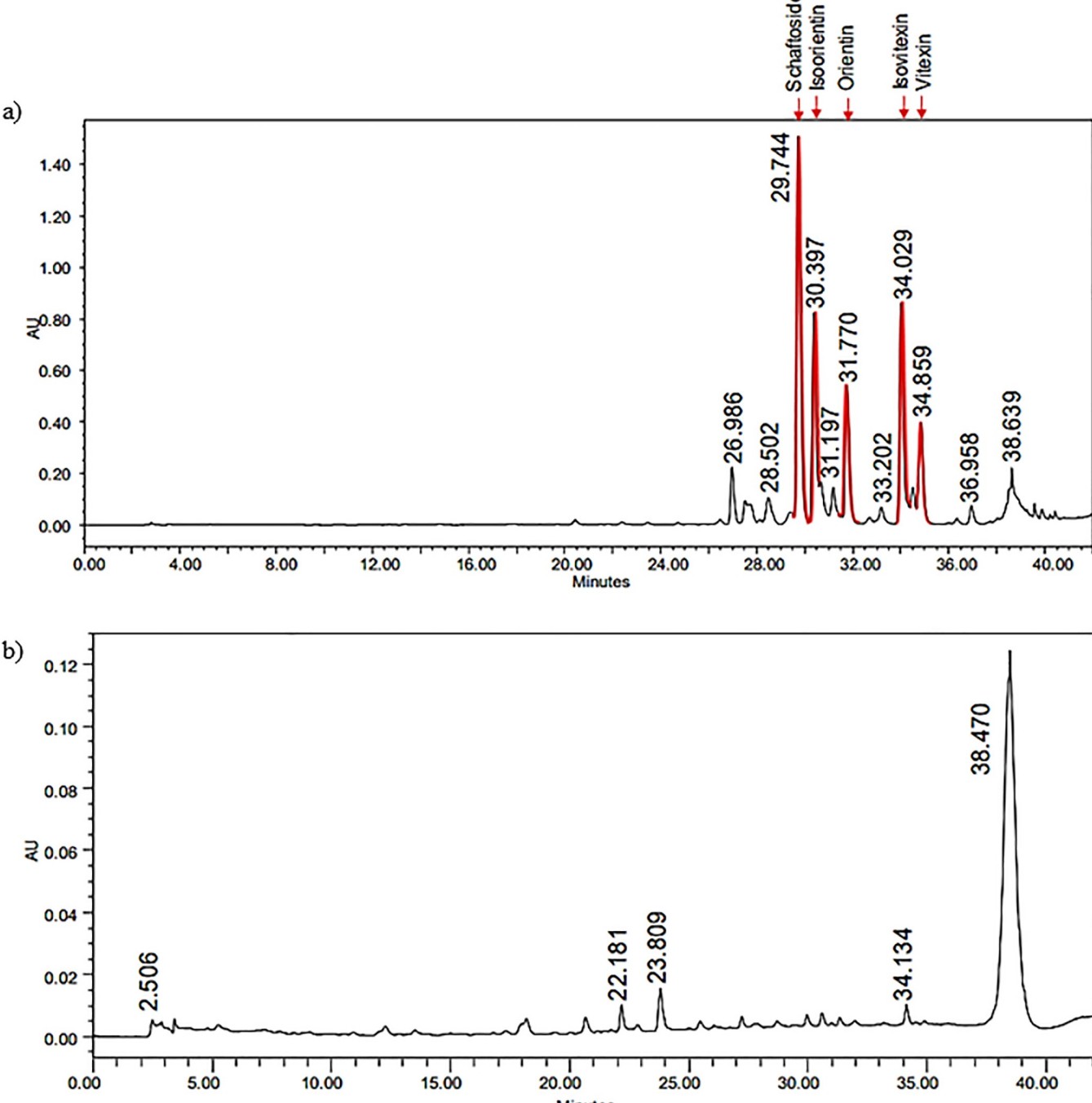

**Fig 4. HPLC chromatogram. a) CN ethanol and b) CN aqueous extracts.**

Natural products remain as a tremendous source of anticancer drug leads or compounds [23]. In 2014, 53% of the new compounds approved for cancer treatment are of natural product origin [24, 25].]. *Clinacanthus nutans* (CN) (Acanthaceae) or Sabah snake grass is a medicinal plant with anticancer effect. The leaves of this plant are consumed as a vegetable or taken as tea because of its common health benefits [26, 27]. It is used as an alternative medicine for

**Table 4. Content of schaftoside, isoorientin, orientin, isovitexin and vitexin in CN ethanol and aqueous extracts.**

| Compound | Average concentration (ppm) | | Average percentage of compound in sample±RSD (*w/w*) | |
|---|---|---|---|---|
| | **CN 100% ethanol** | **CN Aqueous** | **CN 100% ethanol** | **CN Aqueous** |
| Schaftoside | 3654.05 ±4.23 | ND | 0.73±4.23 | ND |
| Isoorientin | 1945.22 ± 4.87 | ND | 0.39±4.87 | ND |
| Orientin | 1204.07± 0.49 | ND | 0.24±0.49 | ND |
| Isovitexin | 1285.24±4.99 | ND | 0.26±4.99 | ND |
| Vitexin | 653.99±1.38 | ND | 0.13±1.38 | ND |

ND: not detected.

Values are means±RSD, n = 3.

insect bites, skin rashes, inflammation, burns and scalds, dysentery, diabetes, herpes infection and cancer [28, 29]. This plant can be found in several South East Asian countries, including Malaysia, Thailand and Indonesia. The vernacular names of CN are "belalai gajah" (Malay), Sabah snake grass (English), you dun cao (Chinese), slaed pang pon (Thai), "dandang gendis" or "kitajam" (Indonesian) [9].

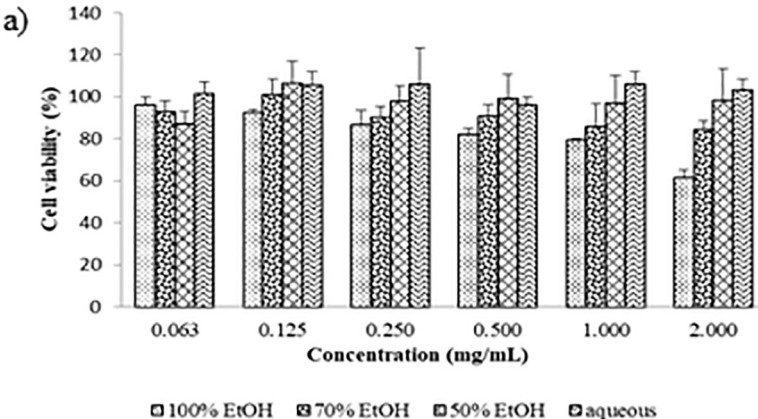

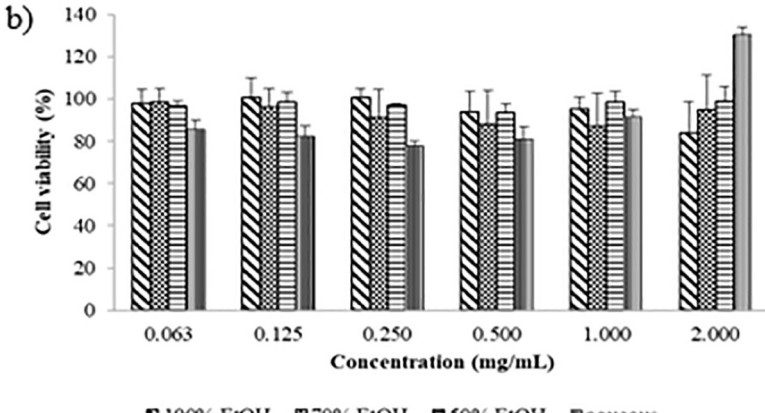

**Fig 5. Cytotoxicity of CN extracts on cells.** (a). Cell viability of CN extracts on human metastatic breast cancer cells, MDA-MB-231 (b). Cell viability of CN extracts in human macrophage-like cells, THP-1. Exposure to CN extracts showed no remarked difference on the cell viability of both types of cells.

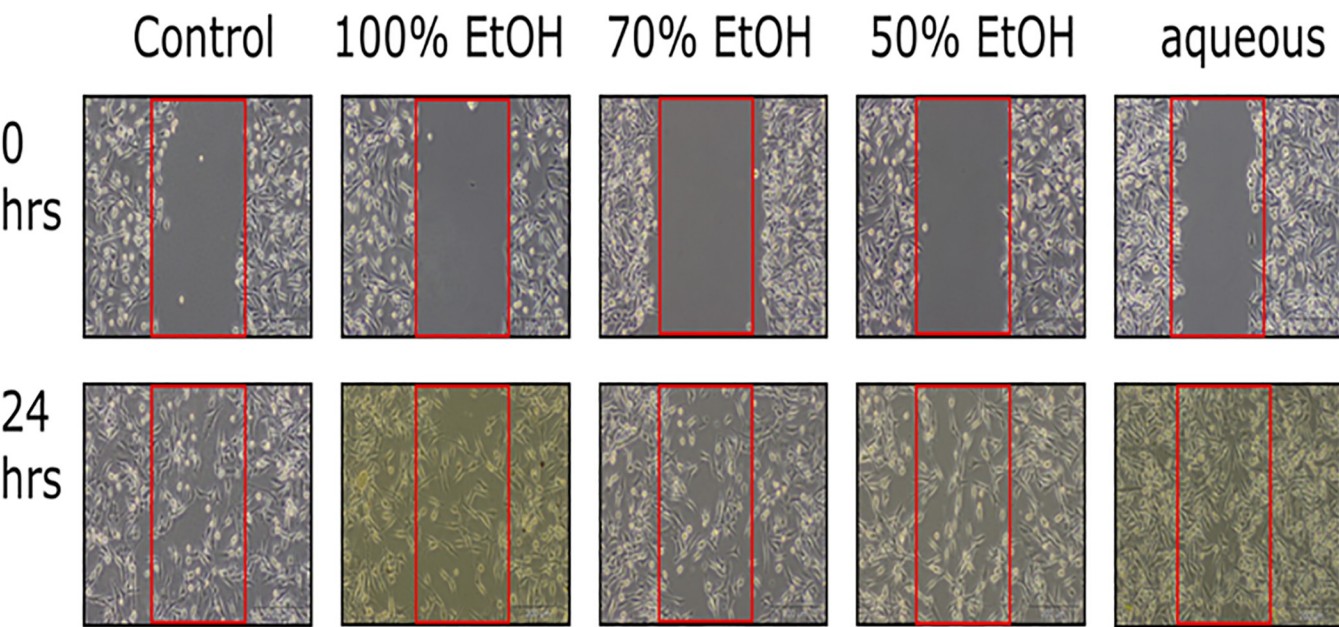

**Fig 6. CN effect on human metastatic breast cancer cells, MDA-MB-231 migration.** Various types of CN extracts cannot inhibit the migration of MDA-MB-231 cells. Cancer cells were able to migrate and closed the scratched wounds in 24 hours.

Traditionally, dried CN leaves were made into tea or blended with green apple in plain water as healthy juice. Water, which is the most polar solvent, can extract the highest total phytochemical content from the leaves [30]. Moreover, the drying temperature plays a critical role in the CN phytochemical constituent extraction, especially total phenolic and flavonoid contents. A study found that the leaf extract from CN has higher saponin, alkaloid, steroid and tannin contents than the stem part. CN plants that are exposed to sunlight during cultivation promote higher phenolic content than plant cultivated under the shade [13].

The central tenet of western medicine is that a drug should be composed of well-known chemical component or a pure single compound that is selective and target-specific [31]. Most of the conventional cytotoxic anticancer drugs were discovered through random high-throughput screening of compounds in cell-based cytotoxicity assays [32]. Thus, the National Cancer Institute under the Developmental Therapeutics Program has developed a panel of 60 human tumor cell lines and adapted as one of the most important steps in evaluating new anti-cancer agent [33]. On the other hand, herbal medicine was dependent on evidence-based approach which accumulated over centuries in the form of traditional medicine or folklore medicine [31, 34]. Herbal medicine comprised of multi-component phytochemicals which identification of its active constituent were difficult [35]. More efforts were needed especially in bioassay experiments to build strong scientific evidences for integration of herbal medicine into mainstream medicine to treat cancer. Perhaps, the achievement of YIV-906 (PHY906), a four-herb formulation to be incorporated as anticancer adjuvant for cancer patients serves as model for other types of herbal plant [36]. In case of CN, elucidation of the its mechanism is of the utmost importance to enable this herbal medicine to be regulated as a drug and differentiate it with the commercial herbal product, hence satisfying the unmet medical need [37].

In this study, the cytotoxic effect of CN was evaluated on human metastatic breast cancer cell line such as MDA-MB-231 cells. Both ethanolic and aqueous CN extracts were not cytotoxic to the MDA-MB-231 cells. Our study was similar to those of Mai et al. [10] and Vajrabaya [38], in which CN was not toxic to mouse macrophage cells, RAW 264.7, mouse fibroblast,

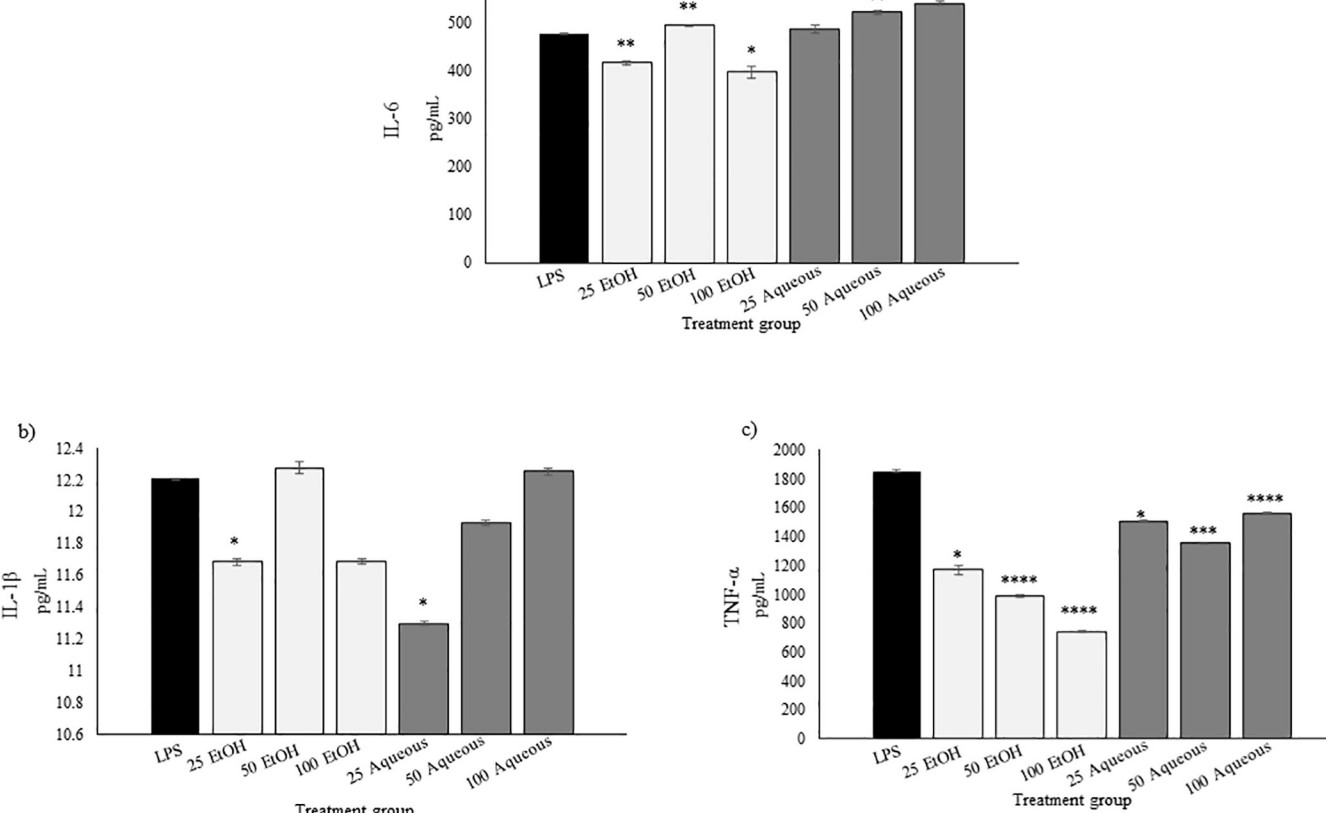

**Fig 7. Effect of CN on IL-6, IL-1β, and TNF-α level of expression in MDA-MB-231/THP-1 co-culture.** a) CN ethanol extract decreased IL-6 expression at 25- and 100 μg/mL but increased its expressions after treatment with CN aqueous extract at 50- and 100 μg/mL. b) CN ethanol extract inhibited the IL-1β expression at 25- and 100 μg/mL but not at 50 μg/mL. CN aqueous extract significantly inhibited IL-1β expression at 25μg/mL c) CN ethanol and aqueous significantly decreased TNF-α expression at all tested concentrations. Data was obtained from three independent experiment replicates and expressed as mean ± SD. $^*p < 0.05$, $^{**}p < 0.01$, $^{***}p < 0.001$ and $^{****}p < 0.0001$ against LPS treatment only.

L929 and human breast cancer cells such as MDA-MB-231 and MCF-7 [11, 39]. However, in other studies, CN exerts antiproliferative effects towards breast cancer cells, MDA-MB-231 [40, 41], ovarian cancer cells, Hela [42], pancreatic ductal adenocarcinoma, AsPC1, BxPC3 and SW1990 [11], erythroleukemia cells and K562 [7, 43] with very potent anticancer activity and $IC_{50}$ less than 30 μg/mL. The difference of antiproliferative activity of CN extract against MDA-MB-231 between this experiment and other studies was due to the type of solvent used to prepare the extracts. Both studies by Mutazah et al. [40] and Quah et al. [41] used methanolic CN extracts but our study used ethanol and distilled water to prepare the CN extracts. In terms of plant part, our study was similar to Quah et al. [41] that used CN leaves, however, Mutazah et al. [40] used bark to prepare CN methanolic extract. Other than that, previous study reported that the harvesting age and harvesting frequencies may influence the phytochemical content in CN plant especially shaftoside, isoorientin and orientin [44]. The highest total phenolic content and flavonoids can be obtained from harvesting at week 16 during first harvest. Other factor that might explained the difference of the antiproliferative activity of CN was that there might be variation in the amount of phytochemical content in various CN extracts. According to a study, phytochemical production in CN plant was high until 6 months of age and decreased until reaching one year of age [45] and increased harvesting frequency results in decreased amount of total phenolic and flavonoids contents in CN plant [44].

Considering that most of the natural products are screened for their antiproliferative activity before being further studied for their anticancer potential, researchers may have missed several potential agents that may not be cytotoxic but can halt carcinogenesis or metastasis through non-cancer cell death-induced mechanism, such as targeting angiogenesis [46], re-educating the immune cells [47] or reducing the smoldering inflammation in the TME [48, 49]. Therefore, the minimum criteria that must be satisfied in the early phase of pre-clinical anticancer drug development process should not be solely based on the cytotoxic parameters, especially for natural product-based agents.

Inflammation is essential to protect the body from virus or bacterial infections. For example, pro-inflammatory endogenous cytokines such as IL-1, TNF-α, and IL-6 are crucial for the resolution of acute inflammation [50]. However, prolonged inflammation or chronic inflammation can positively influence breast cancer growth and progression [5]. High levels of innate cytokines, as evident in chronic inflammation, can promote tumor progression by inducing the sustained activation of NF-κB [51]. The persistent activation of the immune system in chronic inflammation environment enhances genomic lesions and promotes tumor growth [52]. Pleiotropic IL-6 may play a critical role in the communication between cancerous and non-cancerous cells within the TME [53, 54]. IL-6 is a potent cancer cell growth factor that can induce an epithelial–mesenchymal transition phenotype in breast cancer and therapeutic resistance in breast cancer [55, 56]. IL-1β plays a pivotal role in promoting inflammation in TME, angiogenesis and immunosuppression [57]. The pro-inflammatory factors of IL-1 and IL-6 from TAMs facilitate the invasion of cancer cells, and this process might be associated with their receptor up-regulation [58]. These factors are also involved in the Jak-STAT3 signaling pathway in the chronic inflammation-associated tumor growth and immunosuppressive process [58]. The secreted IL-1β into the TME promotes tumorigenesis, tumor invasiveness and immunosuppression [59]. IL-6 is involved in immune evasion process and T-cell-mediated cytotoxicity [60]. Blocking the IL-1β/IL-6 network in TME could halt tumor progression [61].

Our results showed that 25 and 100 μg/mL ethanolic CN extract suppressed the secretion of IL-6 and IL-1β in the co-culture between human TNBC cell line, MDA-MB-231 and human macrophage-like cells such as THP-1 (Fig 7A and 7B). By contrast, CN aqueous extract increased the secretion of IL-6 at 50 and 100 μg/mL but not of IL-1β at 25 and 50 μg/mL. LPS-induced production of pro-inflammatory cytokines were regulated by multiple pathways such as Nrf2/Keap1 and NF-κB pathways thus overlapping modulation of these pathways may have occurred at certain CN concentration as previously shown by other plant phytochemicals [62]. Other than that, the contribution of CN antioxidant effect may have altered the pro-inflammatory cytokine milieu in the co-culture [63, 64]. High IL-6 expression in co-culture experiment may be due to the high basal IL-6 secretion from MDA-MB-231 cells besides production of LPS-induced pro-inflammatory cytokines from THP-1 macrophages [65]. These results suggested that CN extracts were able to ameliorate the inflammation state in the TME via inhibition of IL-6 or IL-1β secretions from the cancer cells and immune cell interactions at certain concentrations.

Tumor necrosis factor alpha (TNF-α) is a pleiotropic cytokine that mainly mediates anti-tumor effects, but this cytokine can also promote tumor progression [66]. The present study showed that the exposure of the CN ethanolic and aqueous extracts suppressed TNF-α cytokine expression. Previous study also showed that various types of CN extracts significantly reduced the LPS-induced TNF-α production of macrophage cells such as RAW 264.7 [10]. The TNF-α involvement in the development of cancer is complex, and the balance between the high and low levels of TNF has an adverse effect on tumor growth [67]. Activated macrophages were the main cells that produced TNF-α in the TME [68]. Studies also showed that

this cytokine was the essential effector cytokines that initiate and maintain chronic inflammation in mouse models [69]. Although TNF-α showed pro-apoptotic functions in tumor cells, most animal models and clinical studies revealed the pro-neoplastic functions of this cytokine [70, 71]. On the other hand, high levels of TNF-α are associated with cancer progression and metastasis [72]. The ability of CN ethanol extract to suppress TNF-α may help to ameliorate the inflamed interaction between breast cancer cells, MDA-MB-231 and macrophage, THP-1.

Breast cancer is not a stand-alone proliferative disease but involved the cooperation between various types of immune cells in the tumor microenvironment [71]. The communications between cancer cells and immune cells were mediated by cytokines by which the cancer cells often take control in shaping the conducive microenvironment for its survival [73, 74]. The conducive microenvironment for cancer cells survival and progression were characterized by the phenomenon of chronic inflammation. Pro-inflammatory cytokines such as IL-1α, IL-1β, IL-6, TNF-α were constitutively expressed in tumor microenvironment to facilitate carcinogenesis [5, 75]. In the current study, CN extracts were not affecting the proliferation and migration of cancer cells but suppressed the pro-inflammatory cytokines such as IL-6, IL-1β, and TNF-α. This showed that amelioration of the pro-inflammatory cytokines in the tumor microenvironment were the key attributes of the anticancer properties of CN extracts.

## Conclusions

In conclusion, CN ethanol and aqueous extracts did not affect the viability of both human breast cancer cells such as MDA-MB-231 and THP-1 macrophage-like cells and was not able to inhibit the migration of cancer cells in vitro. However, the ethanolic CN extracts prevented the smoldering inflammation between cancer cells and immune cells by reducing the level of pro-inflammatory cytokines, such as IL-6, IL-1β and TNF-α. By contrast, the effects of aqueous CN extract on the pro-inflammatory cytokines were limited to IL-1β. Hence, the possible anticancer mechanism through which CN may exert its effect is by reducing the inflammation state of TME. Further research is needed especially on the downstream consequences of the pro-inflammatory cytokines inhibition by CN extracts, as well as the upstream pathways that promote cytokine production such as NF-κB and STATs pathways.

## Acknowledgments

The authors thank the Bioscience Institute, Universiti Putra Malaysia and Toxicology Laboratory, Universiti Kebangsaan Malaysia for experimental facilities.

## Author Contributions

**Conceptualization:** Kok Meng Chan, Endang Kumolosasi, Nor Fadilah Rajab.

**Data curation:** Fariza Juliana Nordin, Lishantini Pearanpan.

**Formal analysis:** Fariza Juliana Nordin, Lishantini Pearanpan.

**Funding acquisition:** Yoke Keong Yong, Khozirah Shaari, Nor Fadilah Rajab.

**Investigation:** Fariza Juliana Nordin, Lishantini Pearanpan.

**Methodology:** Kok Meng Chan, Endang Kumolosasi.

**Project administration:** Yoke Keong Yong, Khozirah Shaari, Nor Fadilah Rajab.

**Supervision:** Kok Meng Chan, Endang Kumolosasi, Nor Fadilah Rajab.

**Writing – original draft:** Fariza Juliana Nordin, Lishantini Pearanpan.

**Writing – review & editing:** Kok Meng Chan, Endang Kumolosasi, Yoke Keong Yong, Nor
  Fadilah Rajab.

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
