## [Decision Letter · Decision Letter 0]

11 Dec 2020

PONE-D-20-27749

Immunomodulatory potentials of Clinacanthus nutans extracts in the co-culture of triple-negative breast cancer cells, MDA-MB-231, and THP-1 macrophages

PLOS ONE

Dear Dr. Rajab,

Thank you for submitting your manuscript to PLOS ONE. After careful consideration, we feel that it has merit but does not fully meet PLOS ONE’s publication criteria as it currently stands. Therefore, we invite you to submit a revised version of the manuscript that addresses the points raised during the review process.

A number of significant issues were raised by the reviewers, one of whom suggested rejection.  Nevertheless, if you feel that you can address these issues we would be willing to consider a revised manuscript. 

We look forward to receiving your revised manuscript.

Kind regards,

Salvatore V Pizzo

Academic Editor

PLOS ONE

Journal Requirements:

2. Please provide additional information about each of the cell lines used in this work, including the source, and any quality control testing procedures (authentication, characterisation, and mycoplasma testing).

For more information, please see http://journals.plos.org/plosone/s/submission-guidelines#loc-cell-lines

Reviewers' comments:

Reviewer's Responses to Questions

**Comments to the Author**

1. Is the manuscript technically sound, and do the data support the conclusions?

Reviewer #1: Partly

Reviewer #2: Partly

2. Has the statistical analysis been performed appropriately and rigorously? 

Reviewer #1: No

Reviewer #2: No

3. Have the authors made all data underlying the findings in their manuscript fully available?

Reviewer #1: Yes

Reviewer #2: No

4. Is the manuscript presented in an intelligible fashion and written in standard English?

Reviewer #1: Yes

Reviewer #2: Yes

5. Review Comments to the Author

Reviewer #1: Kindly mentioned important statistical values inside the abstract.

Further study is needed to fully understand the potential of CN for cancer treatment. Kindly mentioned what further study will be required.

Kindly mentioned source of chemicals and list of equipment. Did you dry the plant?

Please write complete methodology including Plant extraction. This is incomplete. Please include a flow diagram of your extraction procedure.

Did you grind the plant? What is the sieve size?

Is this your new method on Cell culture,Sulforhodamine (SRB) cytotoxicity assay, Migration assay, Co-culture of MDAMB-231/THP-1 macrophage-like cells and Cytokine analysis. Please cite proper reference if you have adapted the methodology.

Please mentioned all abbreviation at the end.

Did you calculate Correlation coefficient, linearity, Limit of detection (LOD) and Limit of quantification (LOQ) for HPLC study. I recommend to include this.

According to your discussion "Our study was similar to

those of Mai et al. 2016 and Khoo et al. 2018, in which CN was not toxic to mouse

macrophage cells, RAW 264.7, mouse fibroblast, L929 [31] and human breast cancer cells

such as MDA-MB-231 and MCF-7 [11, 32]. However, in other studies, CN exerts

antiproliferative effects towards breast cancer cells, MDA-MB-231 [33, 34], ovarian cancer

cells, Hela [35], pancreatic ductal adenocarcinoma, AsPC1, BxPC3 and SW1990 [11],

erythroleukemia cells and K562 [7,36] with very potent anticancer activity and IC50 less than

30 µg/mL."

Some studies showed high activity towards breast caner cell. Why your results does not toxic to mouse

macrophage cells, RAW 264.7, mouse fibroblast, L929 and human breast cancer cells

such as MDA-MB-231 and MCF-7. Please add one paragraph and justify. You have cited the reference but it is not sufficient to proceed. Please compare the methodologies of previous work. You have stated the mechanism as

"Hence, the possible anticancer mechanism through which CN may exert its effect is by

reducing the inflammation state of TME"

If you elaborate this statement in the discussion then your paper will be a better to understand and this is what researchers are looking at.

Please improve your discussion in this context.

Please provide future direction inside the conclusion.

Reviewer #2: Triple negative breast cancer is a difficult to treat phenotype that definitely warrants further investigation. Research into natural ‘local’ remedies to underpin the scientific basis for any effects is interesting and this paper aims to link the two fields by investigating the effects of a well known SE Asian medicinal plant C.Nutans on a triple negative breast cancer cell line and a macrophage cell line in vitro.

Crude extracts from plants contain many phytochemicals, as indicated by the qualitative observation of a wide range of constituents (table 1 ie containing Saponin, Alkaloid, Phenol and Tannin, Flavonoid, Terpenoid, Glycoside, Steroid). HPLC demonstrated a number of flavones and one flavonoid (schaftoside) in the ethanol extract, as may be expected. It isn’t possible to evaluate what the active components are, and whether they could be reproduced by a single purified entity (eg after full characterization by spectral techniques (eg MS, NMR, IR, etc). I am slightly unclear whether this manuscript is about drug discovery (potentially novel compounds from the plant) or herbal medicine, since both are mentioned.

The ethanol and aqueous extracts did not impact on cell viability, and the concept of cancer as a proliferative disease is now old fashioned in the era of immunotherapy. However the authors make a valid point that only screening for agents which inhibit the proliferation of tumour cell lines will miss compounds that effect different pathways. The scratch assay indicated that the extracts did not effect migration either.

It was interesting that the authors included a co-culture experiment to evaluate the effects of the extract on cytokine production. THP-1 are a monocytic leukemia cell line and were differentiated to macrophages using PMA. However, this does not mean they have the characteristics of M2 macrophages (as indicated in the discussion). Other authors consider THP-1 as a good model of M0/M1 differentiation and there are publications using IL4 and IL13 to drive M2 differentiation eg Genin, M., Clement, F., Fattaccioli, A. et al. M1 and M2 macrophages derived from THP-1 cells differentially modulate the response of cancer cells to etoposide. BMC Cancer 15, 577 (2015). https://doi.org/10.1186/s12885-015-1546-9

I think more specific detail on the co-culture model would be valuable, the seeding density of THP-1 on inserts was stated, but the seeding density / confluence of MDA-MB231 triple negative breast cancer cell line was not (just that they were cultured for 6 days). Was the extract (1h in serum free medium) placed in the upper chamber, lower chamber or both? The differentiated THP-1 were then primed with LPS (20ng/ml), to stimulate cytokine production – which is more representative of infection and not cancer. 18h later the culture supernatant was removed for cytokine analyses – was this the upper chamber / lower chamber and do the authors think the cytokine was from the macrophages only or there could be any contribution by the epithelial cell line? What is the rationale for including the epithelial cells in the co-culture experiments – what contribution do the authors think they are making?

The cytokine data was presented as % of activation rather than the absolute values. I think it would be interesting to see the amount of cytokine produced by LPS stimulated THP-1 +/- extracts. It isn’t clear in the methods how many replicates and whether the data produced is normally distributed. If it is, then ANOVA is more appropriate than a simple t test.

Ultimately, the authors state that their extracts are anti-inflammatory on the basis of the changes in production of IL6, IL1β and TNFalpha, although the aqueous extract seems to show a slight increase in IL6 and TNFalpha (above 100%). If there was a meaningful effect, then a dose response should be seen (not evident for IL6 and IL-1β with the ethanol extract).

In terms of the questions asked:

I think there is some key information missing from the methods section (particularly on the co-culture model) which is important to include since this is the basis for the key findings of the paper.

In addition, t-test may not be the most suitable statistical assay since it only compares two means. ANOVA would be more appropriate given the number of variables (concentration, extract).

The cytokine data is oddly presented (% activation) rather than concentration values - which is what I would expect to see.

6. PLOS authors have the option to publish the peer review history of their article (what does this mean?). If published, this will include your full peer review and any attached files.

Reviewer #1: **Yes: **Dr. Muhammad Shahzad Aslam

Reviewer #2: No

---

## [Author Response · Author response to Decision Letter 0]

4 Apr 2021

Reviewer #1:

1. Kindly mentioned important statistical values inside the abstract.

• The statistical values have been added in the abstract.

2. Further study is needed to fully understand the potential of CN for cancer treatment. Kindly mentioned what further study will be required

• The major pathways that link both cancer and inflammation were NF-κB and STATs thus further study on the upstream and downstream pathways is needed to fully understand the mechanism of CN extracts in cooling the inflamed TME in breast cancer. This has been added in the revised manuscript.

3. Kindly mentioned source of chemicals and list of equipment. Did you dry the plant?

• The source of chemicals and equipment have been included in the revised manuscript. Yes, we dried the leaves at room temperature until the leaves is fully dried. 

4. Please write complete methodology including Plant extraction. This is incomplete. Please include a flow diagram of your extraction procedure.

• The methodology section has been added and the flowchart is included in the revised manuscript.

5. Did you grind the plant? What is the sieve size?

• Yes, we grind the leaves into powder and sieve using 315 mm test sieve (Retsch, Germany).

6. Is this your new method on Cell culture,Sulforhodamine (SRB) cytotoxicity assay, Migration assay, Co-culture of MDAMB-231/THP-1 macrophage-like cells and Cytokine analysis. Please cite proper reference if you have adapted the methodology.

The appropriate references have been added in the revised manuscript. 

• Sulforhodamine cytotoxicity assay Holbeck et al. 2010, Orellana et al. 2016. https://doi.org/10.21769/bioprotoc.1984

• Migration assay were carried out according to Wang et al. 2019 https://dx.doi.org/10.2147%2FOTT.S199605

• Co-culture of MDAMB231/THP-1 macrophage-like cells experiment was carried out according to previous study by Smith 2015 https://doi.org/10.21769/bioprotoc.1638 and Mai 2016 https://doi.org/10.3389/fphar.2016.00007

• Schematic diagram of the method has been added in the revised manuscript.

• Cytokine analysis were carried out as per manufacturer protocols. 

7. Please mentioned all abbreviation at the end.

• The abbreviation has been added in the revised manuscript.

8. Did you calculate Correlation coefficient, linearity, Limit of detection (LOD) and Limit of quantification (LOQ) for HPLC study. I recommend to include this.

• Our CN extracts were sent for HPLC service to Forest Research Institute of Malaysia (FRIM). The institution used in-house HPLC validation method which they cannot make the HPLC validation methods available to their clients. However, the chromatogram of the standards used in the experiment has been added in the revised manuscript. 

9. According to your discussion "Our study was similar to

those of Mai et al. 2016 and Khoo et al. 2018, in which CN was not toxic to mouse

macrophage cells, RAW 264.7, mouse fibroblast, L929 [31] and human breast cancer cells such as MDA-MB-231 and MCF-7 [11, 32]. However, in other studies, CN exerts

antiproliferative effects towards breast cancer cells, MDA-MB-231 [33, 34], ovarian cancer cells, Hela [35], pancreatic ductal adenocarcinoma, AsPC1, BxPC3 and SW1990 [11], erythroleukemia cells and K562 [7,36] with very potent anticancer activity and IC50 less than 30 µg/mL."

Some studies showed high activity towards breast cancer cell. Why your results does not toxic to mouse macrophage cells, RAW 264.7, mouse fibroblast, L929 and human breast cancer cells such as MDA-MB-231 and MCF-7. Please add one paragraph and justify. You have cited the reference but it is not sufficient to proceed. Please compare the methodologies of previous work.

• The results justification and method comparison has been added in the discussion section. 

“In this study, the cytotoxic effect of CN was evaluated on human metastatic breast cancer cell line such as MDA-MB-231 cells. Both ethanolic and aqueous CN extracts were not cytotoxic to the MDA-MB-231 cells. Our study was similar to those of Mai et al. [10] and Vajrabaya [39], in which CN was not toxic to mouse macrophage cells, RAW 264.7, mouse fibroblast, L929 and human breast cancer cells such as MDA-MB-231 and MCF-7 [11, 40]. However, in other studies, CN exerts antiproliferative effects towards breast cancer cells, MDA-MB-231 [41, 42], ovarian cancer cells, Hela [43], pancreatic ductal adenocarcinoma, AsPC1, BxPC3 and SW1990 [11], erythroleukemia cells and K562 [7,44] with very potent anticancer activity and IC50 less than 30 µg/mL. The difference of antiproliferative activity of CN extract against MDA-MB-231 between this experiment and other studies was due to the type of solvent used to prepare the extracts. Both studies by Mutazah et al. [41] and Quah et al. [42] used methanolic CN extracts but our study used ethanol and distilled water to prepare the CN extracts. In terms of plant part, our study was similar to Quah et al. [42] that used CN leaves, however, Mutazah et al. [41] used bark to prepare CN methanolic extract. Other than that, previous study reported that the harvesting age and harvesting frequencies may influence the phytochemical content in CN plant especially shaftoside, isoorientin and orientin [45]. The highest total phenolic content and flavonoids can be obtained from harvesting at week 16 during first harvest. Other factor that might explained the difference of the antiproliferative activity of CN was that there might be variation in the amount of phytochemical content in various CN extracts. According to a study, phytochemical production in CN plant was high until 6 months of age and decreased until reaching one year of age [46] and increased harvesting frequency results in decreased amount of total phenolic and flavonoids contents in CN plant [45]”

10. You have stated the mechanism as, "Hence, the possible anticancer mechanism through which CN may exert its effect is by reducing the inflammation state of TME" 

If you elaborate this statement in the discussion then your paper will be a better to understand and this is what researchers are looking at.

Please improve your discussion in this context.

• The discussion on CN mechanism has been added in the revised manuscript. 

“Breast cancer is not a stand-alone proliferative disease but involved the cooperation between various types of immune cells in the tumor microenvironment [76]. The communications between cancer cells and immune cells were mediated by cytokines by which the cancer cells often take control in shaping the conducive microenvironment for its survival [77, 78]. The conducive microenvironment for cancer cells survival and progression were characterized by the phenomenon of chronic inflammation. Pro-inflammatory cytokines such as IL-1α, IL-1β, IL-6, TNF-α were constitutively expressed in tumor microenvironment to facilitate carcinogenesis [5, 79]. In the current study, CN extracts were not affecting the proliferation and migration of cancer cells but suppressed the pro-inflammatory cytokines such as IL-6, IL-1β, and TNF-α. This showed that amelioration of the pro-inflammatory cytokines in the tumor microenvironment were the key attributes of the anticancer properties of CN extracts.”

11. Please provide future direction inside the conclusion.

• Future direction of the study has been added in the conclusion section.

“Further research is needed especially on the downstream consequences of the pro-inflammatory cytokines inhibition by CN extracts, as well as the upstream pathways that promote cytokine production such as NF-κB and STATs pathways.”

Reviewer #2:

1. Triple negative breast cancer is a difficult to treat phenotype that definitely warrants further investigation. Research into natural ‘local’ remedies to underpin the scientific basis for any effects is interesting and this paper aims to link the two fields by investigating the effects of a well-known SE Asian medicinal plant C.nutans on a triple negative breast cancer cell line and a macrophage cell line in vitro.

Crude extracts from plants contain many phytochemicals, as indicated by the qualitative observation of a wide range of constituents (table 1 ie containing Saponin, Alkaloid, Phenol and Tannin, Flavonoid, Terpenoid, Glycoside, Steroid). HPLC demonstrated a number of flavones and one flavonoid (schaftoside) in the ethanol extract, as may be expected. It isn’t possible to evaluate what the active components are, and whether they could be reproduced by a single purified entity (e.g., after full characterization by spectral techniques (eg MS, NMR, IR, etc). I am slightly unclear whether this manuscript is about drug discovery (potentially novel compounds from the plant) or herbal medicine, since both are mentioned.

• It is about herbal medicine. This plant is widely used as alternative medicine for cancer especially in Malaysia, we sought to investigate the scientific evidence of CN extracts effect in breast cancer. We have added a few lines in the discussion part to clarify it. 

“The central tenet of western medicine is that a drug should be composed of well-known chemical component or a pure single compound that is selective and target-specific [32]. Most of the conventional cytotoxic anticancer drugs were discovered through random high-throughput screening of compounds in cell-based cytotoxicity assays [33]. Thus, the National Cancer Institute under the Developmental Therapeutics Program has developed a panel of 60 human tumor cell lines and adapted as one of the most important steps in evaluating new anticancer agent [34]. On the other hand, herbal medicine was dependent on evidence-based approach which accumulated over centuries in the form of traditional medicine or folklore medicine [32, 35]. Herbal medicine comprised of multi-component phytochemicals which identification of its active constituent were difficult [36]. More efforts were needed especially in bioassay experiments to build strong scientific evidences for integration of herbal medicine into mainstream medicine to treat breast cancer. Perhaps, the achievement of YIV-906 (PHY906), a four-herb formulation to be incorporated as anticancer adjuvant for cancer patients serves as model for other types of herbal plant [37]. In case of CN, elucidation of the its mechanism is of the utmost importance to enable this herbal medicine to be regulated as a drug and differentiate it with the commercial herbal product, hence satisfying the unmet medical need [38]”.

2. The ethanol and aqueous extracts did not impact on cell viability, and the concept of cancer as a proliferative disease is now old fashioned in the era of immunotherapy. However, the authors make a valid point that only screening for agents which inhibit the proliferation of tumour cell lines will miss compounds that effect different pathways. The scratch assay indicated that the extracts did not affect migration either.

It was interesting that the authors included a co-culture experiment to evaluate the effects of the extract on cytokine production. THP-1 are a monocytic leukemia cell line and were differentiated to macrophages using PMA. However, this does not mean they have the characteristics of M2 macrophages (as indicated in the discussion). Other authors consider THP-1 as a good model of M0/M1 differentiation and there are publications using IL4 and IL13 to drive M2 differentiation eg Genin, M., Clement, F., Fattaccioli, A. et al. M1 and M2 macrophages derived from THP-1 cells differentially modulate the response of cancer cells to etoposide. BMC Cancer 15, 577 (2015). https://doi.org/10.1186/s12885-015-1546-9

I think more specific detail on the co-culture model would be valuable, the seeding density of THP-1 on inserts was stated, but the seeding density / confluence of MDA-MB231 triple negative breast cancer cell line was not (just that they were cultured for 6 days). 

• MDA-MB-231 cells were seeded at 2x105 cells/well a day before THP-1 incubation with PMA ended. 

3. Was the extract (1h in serum free medium) placed in the upper chamber, lower chamber or both? 

• CN extracts were added into both upper and lower chambers. 

4. The differentiated THP-1 were then primed with LPS (20ng/ml), to stimulate cytokine production – which is more representative of infection and not cancer. 

• Persistent inflammation can contribute to carcinogenesis. Interaction between cancer cells and immune cells can increased the pro-inflammatory cytokines in the tumor microenvironment. In this experiment, LPS were added to activate THP-1 macrophages and stimulate pro-inflammatory cytokines productions thus created inflamed microenvironment in vitro. The inflamed microenvironment mimics the chronic inflammation that occurs in cancer disease. 

5. 18h later the culture supernatant was removed for cytokine analyses – was this the upper chamber / lower chamber and do the authors thing the cytokine was from the macrophages only or there could be any contribution by the epithelial cell line? 

• After 18 hours, culture supernatant from upper and lower chambers were combined and concentrated using Amicon® Ultra-4 centrifugal filter units (Merck). Yes, the epithelial cell line can contribute to the cytokine production. MDA-MB-231 can produced IL-6. 

6. What is the rationale for including the epithelial cells in the co-culture experiments – what contribution do the authors think they are making?

• The rationale for including the epithelial cells in the co-culture experiments was to mimic the tumor microenvironment which consist of cancer cells and various types of immune cells such as macrophages. THP-1 macrophages were used to mimic the macrophages in tumor microenvironment. Interactions between cancer cells and macrophages were mediated by the pro- inflammatory cytokines such as IL-1β, IL-6 and TNF-α. These pro-inflammatory cytokines create inflamed microenvironment which support cancer progression. 

7. The cytokine data was presented as % of activation rather than the absolute values. I think it would be interesting to see the amount of cytokine produced by LPS stimulated THP-1 +/- extracts. It isn’t clear in the methods how many replicates and whether the data produced is normally distributed. If it is, then ANOVA is more appropriate than a simple t test.

• All data were normally distributed as analyzed using Shapiro-Wilks test. Three different replicates were done. We have reanalysed using ANOVA for IL-1β cytokine data. However, IL-6 and TNF-α data violated the homogeneity of variance (Levene’s test) thus Welch test with Games-Howell post-hoc were carried out. All changes have been included in the revised manuscript.

8. Ultimately, the authors state that their extracts are anti-inflammatory on the basis of the changes in production of IL6, IL1β and TNF�, although the aqueous extract seems to show a slight increase in IL6 and TNF� (above 100%). If there was a meaningful effect, then a dose response should be seen (not evident for IL6 and IL-1β with the ethanol extract).

• CN ethanol extract suppressed IL-6 and IL-1β at 25- and 100 µg/mL. However, CN aqueous extract does not suppress IL-6 but IL-1β at 25 µg/mL. The biphasic cytokine suppression effects occurred at low and high concentrations may be due to overlapping modulation of antioxidant and cytokine inhibitory pathway as reported by many phytochemical based anticancer agents [Jodynis-Liebert & Kujawska 2020. https://doi.org/10.3390/jcm9030718]. Oxidative stress is regulated by Nrf2/Keap1 and NF-κB that is also a redox-regulated transcription factor which also regulates inflammatory response [Speciale et al. 2011. https://doi.org/10.2174/156652411798062395]. As antioxidant activity of CN extracts has been reported by previous studies thus CN extracts may have affected one or both pathways that results in the biphasic dose-response [Che Sulaiman et al. 2015. https://doi.org/10.5897/AJPP2015.4396].

9. I think there is some key information missing from the methods section (particularly on the co-culture model) which is important to include since this is the basis for the key findings of the paper.

• We have amended the co-culture method section and added a schematic diagram of co-culture experiment for better understanding of the method. 

10. In addition, t-test may not be the most suitable statistical assay since it only compares two means. ANOVA would be more appropriate given the number of variables (concentration, extract).

• We have reanalyzed the data using ANOVA and Tukey’s post hoc test for IL-1β cytokine data. Welch F test with Games-Howell post hoc analysis were done on IL-6 and TNF-α. These changes have been added in the revised manuscript.

11. The cytokine data is oddly presented (% activation) rather than concentration values - which is what I would expect to see.

• We have changed the cytokine data to concentration values (pg/mL). This change is included in the revised manuscript. The revised cytokine graphs have been added in the revised manuscript.

---

## [Decision Letter · Decision Letter 1]

24 May 2021

PONE-D-20-27749R1

Immunomodulatory potentials of *Clinacanthus nutans* extracts in the co-culture of triple-negative breast cancer cells, MDA-MB-231, and THP-1 macrophages

PLOS ONE

Dear Dr. Rajab,

Thank you for submitting your manuscript to PLOS ONE. After careful consideration, we feel that it has merit but does not fully meet PLOS ONE’s publication criteria as it currently stands. Therefore, we invite you to submit a revised version of the manuscript that addresses the points raised during the review process.

Several significant issues remains with respect to your manuscript. Reviewer 2 is concerned on two points, namely, can a “three” point curve really be called “biphasic”? and second, have the authors over interpreted their data with respect to actual in vivo tumors?

We look forward to receiving your revised manuscript.

Kind regards,

Salvatore V Pizzo

Academic Editor

PLOS ONE

Reviewers' comments:

Reviewer's Responses to Questions

**Comments to the Author**

1. If the authors have adequately addressed your comments raised in a previous round of review and you feel that this manuscript is now acceptable for publication, you may indicate that here to bypass the “Comments to the Author” section, enter your conflict of interest statement in the “Confidential to Editor” section, and submit your "Accept" recommendation.

Reviewer #1: All comments have been addressed

Reviewer #2: All comments have been addressed

Reviewer #3: (No Response)

2. Is the manuscript technically sound, and do the data support the conclusions?

Reviewer #1: Yes

Reviewer #2: Partly

Reviewer #3: Yes

3. Has the statistical analysis been performed appropriately and rigorously? 

Reviewer #1: Yes

Reviewer #2: Yes

Reviewer #3: Yes

4. Have the authors made all data underlying the findings in their manuscript fully available?

Reviewer #1: Yes

Reviewer #2: Yes

Reviewer #3: Yes

5. Is the manuscript presented in an intelligible fashion and written in standard English?

Reviewer #1: Yes

Reviewer #2: Yes

Reviewer #3: Yes

6. Review Comments to the Author

Reviewer #1: Please check all reference before final submission. It is also recommended to avoid any literature that is retracted.

Reviewer #2: CN is an interesting plant and it’s ethnobotanic historic uses certainly recommend further investigation of it’s medicinal properties.

The authors isolate phytochemicals from CN leaves using ethanol / aqueous methods and partially characterize their contents using a range of qualitative tests plus HPLC. The extracts are not cytotoxic and do not induce migration in a scratch assay.

It is an interesting observation that 1h pre-treatment with CN extracts can affect LPS-induced pro-inflammatory cytokine production from human cancer cell lines ie THP-1 macrophage / MDA-MB-321 epithelial co cultures in vitro.

The authors demonstrated that ethanol extracts decreased IL6 production at 25µg/ml and 100µg/ml but not 50µg/ml, however IL6 levels were significantly *increased* with 50µg/ml and 100 µg/ml of aqueous extracts. A similar mixed pattern was seen with IL-1β production in response to the ethanol extracts (ie decreased at 25µg/ml and 100µg/ml but not 50µg/ml). Aqueous extracts inhibited IL-1β production at the two lower concentrations. CN extracts (both ethanolic and aqueous) significantly inhibited TNF alpha production at all the concentrations tested. I presume that the assay conditions were optimized, but it would be interesting to see dose responses over time. Both cell lines are known to express TLR4, which I guess justifies the use of LPS in this model. I am not sure 3 data points (low, high, low) can be reliably called ‘biphasic’ without further information.

Whilst it is good to mimic a tumor micro-environment in vitro, care has to be taken when extrapolating findings and this manuscript has gone too far in interpreting these observations as representing an 'anti-cancer' effect. I disagree that cultured THP-1 represent M2 macrophages, and they are considered more like M1 in the literature (as previously stated).

Reviewer #3: The article by Nordin et al evaluates the effects of Clinacanthus nutans on the co-culture of MDA-MB-231 cancer cells and human THP-1 macrophages, by looking at its effects on the viability, migratory and inflammatory milieu of the co-culture.

The manuscript is well-written and conclusions are drawn appropriately given the supporting evidence.

The authors have responded appropriately to the recommended revisions given by the previous two reviewers.

Minor comments:

• Introduction section “the leaves of this plant are commonly used as water decoction for oral ingestion or soaked in alcohol for topical application to the affected area [9]”

‘the affected area’ should have been clarified.

7. PLOS authors have the option to publish the peer review history of their article (what does this mean?). If published, this will include your full peer review and any attached files.

Reviewer #1: **Yes: **Muhammad Shahzad Aslam

Reviewer #2: No

Reviewer #3: **Yes: **Liyana Ahmad

---

## [Author Response · Author response to Decision Letter 1]

1 Jul 2021

Notes for reviewer’s comments are as below and included in the uploaded revised manuscript. 

Reviewer #1:

1. Please check all reference before final submission. It is also recommended to avoid any literature that is retracted.

• Authors have checked all references and make sure that there is no retracted literature included in the manuscript to date. Five references have been removed and 2 newly added references has been updated in the list.

Reviewer #2:

1. CN is an interesting plant and it’s ethnobotanic historic uses certainly recommend further investigation of it’s medicinal properties.

The authors isolate phytochemicals from CN leaves using ethanol / aqueous methods and partially characterize their contents using a range of qualitative tests plus HPLC. The extracts are not cytotoxic and do not induce migration in a scratch assay.

It is an interesting observation that 1h pre-treatment with CN extracts can affect LPS-induced pro-inflammatory cytokine production from human cancer cell lines ie THP-1 macrophage / MDA-MB-321 epithelial co cultures in vitro.

• Yes. Similar observations can be obtained from the study by Mai et al. 2016 (https://dx.doi.org/10.3389%2Ffphar.2016.00007). They showed that an hour pre-treatment with CN extract can affect the pro-inflammatory cytokines expression in mouse macrophages RAW 264.7 cells. 

2. The authors demonstrated that ethanol extracts decreased IL6 production at 25µg/ml and 100µg/ml but not 50µg/ml, however IL6 levels were significantly *increased* with 50µg/ml and 100 µg/ml of aqueous extracts. A similar mixed pattern was seen with IL-1β production in response to the ethanol extracts (ie decreased at 25µg/ml and 100µg/ml but not 50µg/ml). Aqueous extracts inhibited IL-1β production at the two lower concentrations. CN extracts (both ethanolic and aqueous) significantly inhibited TNF alpha production at all the concentrations tested. I presume that the assay conditions were optimized, but it would be interesting to see dose responses over time. Both cell lines are known to express TLR4, which I guess justifies the use of LPS in this model. I am not sure 3 data points (low, high, low) can be reliably called ‘biphasic’ without further information.

• Authors have removed the biphasic effect from the discussion part. The corrections have been included in the revised manuscript.

“Our results showed that 25 and 100 µg/mL ethanolic CN extract suppressed the secretion of IL-6 and IL-1β in the co-culture between human TNBC cell line, MDA-MB-231 and human macrophage-like cells such as THP-1 (Figure 7a and 7b). By contrast, CN aqueous extract increased the secretion of IL-6 at 50 and 100 µg/mL but not of IL-1β at 25 and 50 µg/mL. LPS-induced production of pro-inflammatory cytokines were regulated by multiple pathways such as Nrf2/Keap1 and NF-κB pathways thus overlapping modulation of these pathways may have occurred at certain CN concentration as previously shown by other plant phytochemicals [62]. Other than that, the contribution of CN antioxidant effect may have altered the pro-inflammatory cytokine milieu in the co-culture [63,64]. High IL-6 expression in co-culture experiment may be due to the high basal IL-6 secretion from MDA-MB-231 cells besides production of LPS-induced pro-inflammatory cytokines from THP-1 macrophages [65]. These results suggested that CN extracts were able to ameliorate the inflammation state in the TME via inhibition of IL-6 or IL-1β secretions from the cancer cells and immune cell interactions at certain concentrations.”

3. Whilst it is good to mimic a tumor micro-environment in vitro, care has to be taken when extrapolating findings and this manuscript has gone too far in interpreting these observations as representing an 'anti-cancer' effect. I disagree that cultured THP-1 represent M2 macrophages, and they are considered more like M1 in the literature (as previously stated).

• Authors agreed to amend the interpretation of the results. Instead of emphasizing on the anticancer effect, we rewrite the discussion part to suggest that CN ameliorates the inflammation condition between cancer cells and macrophages. The appropriate corrections have been included in the revised manuscript. 

Reviewer #3:

The article by Nordin et al evaluates the effects of Clinacanthus nutans on the co-culture of MDA-MB-231 cancer cells and human THP-1 macrophages, by looking at its effects on the viability, migratory and inflammatory milieu of the co-culture.

The manuscript is well-written and conclusions are drawn appropriately given the supporting evidence.

The authors have responded appropriately to the recommended revisions given by the previous two reviewers.

Minor comments:

Introduction section “the leaves of this plant are commonly used as water decoction for oral ingestion or soaked in alcohol for topical application to the affected area [9]”

‘the affected area’ should have been clarified.

• Authors decided to amend the phrase, “soaked in alcohol for topical application to the affected area [9]” as it is related to herpes related diseases remedy. We rephrased the statement as “As anticancer remedies, the leaves of this plant are commonly used as water decoction for oral ingestion”.

---

## [Editor Report · Decision Letter 2]

29 Jul 2021

Immunomodulatory potentials of *Clinacanthus nutans* extracts in the co-culture of triple-negative breast cancer cells, MDA-MB-231, and THP-1 macrophages

PONE-D-20-27749R2

Dear Dr. Rajab,

We’re pleased to inform you that your manuscript has been judged scientifically suitable for publication and will be formally accepted for publication once it meets all outstanding technical requirements.

Kind regards,

Salvatore V Pizzo

Academic Editor

PLOS ONE
---

## [Editor Report · Acceptance letter]

2 Aug 2021

PONE-D-20-27749R2 

Immunomodulatory potential of *Clinacanthus nutans* extracts in the co-culture of triple-negative breast cancer cells, MDA-MB-231, and THP-1 macrophages. 

Dear Dr. Rajab:

I'm pleased to inform you that your manuscript has been deemed suitable for publication in PLOS ONE. Congratulations! Your manuscript is now with our production department. 

Kind regards, 

on behalf of

Dr. Salvatore V Pizzo 

Academic Editor

PLOS ONE